# Galectin 1—A Key Player between Tissue Repair and Fibrosis

**DOI:** 10.3390/ijms23105548

**Published:** 2022-05-16

**Authors:** Anca Hermenean, Daniela Oatis, Hildegard Herman, Alina Ciceu, Giovanbattista D’Amico, Maria Consiglia Trotta

**Affiliations:** 1Faculty of Medicine, Vasile Goldis Western University of Arad, 310414 Arad, Romania; danielaoatis@gmail.com; 2“Aurel Ardelean” Institute of Life Sciences, Vasile Goldis Western University of Arad, 310414 Arad, Romania; hildegard.i.herman@gmail.com (H.H.); alinaciceu80@gmail.com (A.C.); damicomichele@hotmail.it (G.D.); 3Department of Experimental Medicine, University of Campania “Luigi Vanvitelli”, 80138 Naples, Italy; mariaconsiglia.trotta2@unicampania.it

**Keywords:** galectin 1, wound healing, fibrosis, diabetic retinopathy, diabetic nephropathy, liver fibrosis, pancreatic fibrosis, idiopathic pulmonary fibrosis

## Abstract

Galectins are ten family members of carbohydrate-binding proteins with a high affinity for β galactose-containing oligosaccharides. Galectin-1 (Gal-1) is the first protein discovered in the family, expressed in many sites under normal and pathological conditions. In the first part of the review article, we described recent advances in the Gal-1 modulatory role on wound healing, by focusing on the different phases triggered by Gal-1, such as inflammation, proliferation, tissue repair and re-epithelialization. On the contrary, Gal-1 persistent over-expression enhances angiogenesis and extracellular matrix (ECM) production via PI3K/Akt pathway activation and leads to keloid tissue. Therefore, the targeted Gal-1 modulation should be considered a method of choice to treat wound healing and avoid keloid formation. In the second part of the review article, we discuss studies clarifying the role of Gal-1 in the pathogenesis of proliferative diabetic retinopathy, liver, renal, pancreatic and pulmonary fibrosis. This evidence suggests that Gal-1 may become a biomarker for the diagnosis and prognosis of tissue fibrosis and a promising molecular target for the development of new and original therapeutic tools to treat fibrosis in different chronic diseases.

## 1. Introduction

Galectins are lectins with a highly conserved carbohydrate recognition domain (CRD) and affinity for β galactose-containing oligosaccharides [1]. The family includes 10 members with a large tissue distribution, of which some have a particular specificity [2]. Biochemical studies report their solubility, lack of transmembrane domain, and location mainly in the cytoplasm and less in the nucleus. Galectins have specificity for glycoproteins, but sometimes different chains of a single protein can be recognized by different lectins [3]. The biological activities may be exerted both intracellularly and extracellularly, based on their ability to recognize multiple ligands, being involved mainly in cancer, immunity and inflammation [1,4,5].

Galectin-1 (Gal-1) is the first protein discovered in the family, as a homodimer of 14-kDa subunits, folding in a sandwich of two anti-parallel β-sheets, with two galactoside-binding sites, and is expressed in many sites under normal and pathological conditions [6]. In the cell, Gal-1 is localized within the nucleus, cytoplasm, on the cell surface, and in the extracellular matrix (ECM), where it is secreted [6]. Gal-1 plays a role in a variety of cell functions including proliferation, migration and adhesion, cell growth, immune responses, apoptosis, inflammation, intercellular and cell–matrix interaction and carcinogenesis [7,8,9].

## 2. Galectin 1 in Wound Healing

Wound healing is a complex physiological process that includes hemostasis, inflammation, proliferation and repair and remodeling [10]. The hemostasis is the first stage which occurs immediately after injury with the formation of a provisional wound matrix. Furthermore, the inflammatory phase of the healing starts with neutrophil recruitment, followed by macrophages replacement and phagocytosis [11,12], as well as by the secretion of growth factors and cytokines, promoting cell proliferation and synthesis of ECM components [10,13]. The re-epithelialization process is ensured by local keratinocytes at the wound edges and by stem cells [14,15]. The restoration of the skin vascular networking is a complex cascade event promoted by growth factors, e.g., vascular endothelial growth factor (VEGF) and platelet-derived growth factor (PDGF) [10]. The last step in the proliferation phase is the development of the acute granulation tissue, characterized by macrophages, fibroblasts, capillaries, collagen and blood vessels. The fibroblasts are very active in producing collagen and ECM components, being an important step to provide support for cell adhesion and re-epithelization [16,17]. Finally, the number of fibroblasts is reduced by myofibroblast differentiation and they undergo apoptosis [18]. Moreover, the ECM replaces collagen III, produced in the proliferative phase, with a stronger, collagen I (Col-I) [10]. Further, the myofibroblasts induce wound contractions and contribute to a decrease in the developing scar [19].

### 2.1. Hemostasis and Platelet Adhesion/Aggregation

Platelet adhesion to the ECM at vascular injury sites represents the early steps to stop the bleeding. This process primarily involves binding to fibrillar collagen, fibronectin and laminin [20]. Gal-1 induces the conformational changes of the α_IIb_β3-integrin receptors on the platelet surface and allows fibrinogen binding, leading to the aggregation of platelets into a hemostatic plug [21]. Gal-1 binds to human platelets in a carbohydrate-dependent manner, suggesting that it might play key role in the hemostatic process [22]. Gal-1 binds to α_IIb_β3 integrin and forms lattices that induce integrin clustering and lead to platelet activation [20]. The Gal-1-induced platelet activation involves Ca^2+^ mobilization, phosphorylation of mitogen-activated protein kinases (MAPKs), Akt and β3 integrin [23,24]. In addition, it was demonstrated that not just the soluble Gal-1 is involved in this process. The immobilized Gal-1 is an efficient substrate for platelet adhesion, formation of their filopodia and lamellipodia [23]. Recent studies suggest that the platelet aggregation could be potentiated by both Gal-1 and platelet factor 4 (CXCL4), with supportive effects [25]. The primary hemostasis evaluated in Lgals1^−/−^ and WT mice, shows Gal-1 deficiency in prolonged bleeding time and may be considered a consequence of Gal-1 deficiency in both endothelial cells and/or platelets [23].

### 2.2. Inflammation

Gal-1 exerts immune regulatory activities in animal models of acute/chronic inflammation and plays a role in the repair of injured tissue. Different studies evidenced immunomodulatory functions of Gal-1, and due to its inhibitory effects on neutrophil and T cell trafficking and induction of T cell apoptosis [26,27,28,29,30], may have anti-inflammatory effects. Moreover, a pro-resolving role of Gal-1 in acute inflammation has been also suggested due to its ability to induce phosphatidylserine (PS) exposure on the membrane surface of neutrophils on in vitro studies [31]. Recently, Law et al. [32] demonstrated the anti-trafficking role of endogenous Gal-1 and the pro-apoptotic function of exogenous soluble Gal-1, which is critical for resolving inflammation and tissue repair.

During the resolution of acute inflammation, macrophages undergo reprogramming from pro-inflammatory phenotype (M1) to anti-inflammatory or reparative phenotype (M2) [33,34,35,36]. This pro-resolving switch of macrophages is facilitated by beta interferon (IFN-β), contributing to the resolution of inflammation and healing [37]. Interestingly, Gal-1 was able to facilitate macrophage reprogramming into M2 phenotype and resolution of inflammation through IFN-β [38].

### 2.3. Proliferation, Tissue Repair and Re-Epithelialization

Myofibroblasts activation has a key role in wound healing, including extracellular matrix synthesis, growth factor synthesis and angiogenesis [39,40]. Interestingly, it was noticed that Gal-1 induced myofibroblast activation, migration, and proliferation by upregulation of reactive oxygen species (ROS)-producing protein, nicotinamide adenine dinucleotide phosphate oxidase (NADPH) oxidase 4 (NOX4), through the neuropilin-1/Smad3 signaling pathway in myofibroblasts [41]. In addition, Gal-1 is expressed in skin keratinocytes and mediates matrix interactions, suggesting a potential role in re-epithelialization during wound healing [42]. Gal-1 is upregulated during the early phases of healing of rat skin and tracheal tissue [43,44], while subcutaneous injection of Gal-1 into wound areas accelerated the healing of normal and diabetic wounds [41], suggesting its role in wound repair. However, a previous study showed that both endogenous and exogenous Gal-1 did not influence the re-epithelialization rate of corneal wounds [45].

Moreover, Gal-1 seems to play an important role in controlling neovascularization [46,47]. Gal-1 had pro-angiogenic properties by binding to neuropilin (NRP)-1 receptor to induce angiogenesis, vascular permeability, and wound-healing [48], while loss of endogenous Gal-1 in endothelial cells results in impaired angiogenesis [49,50].

Overall, the schematic involvement of Gal-1 in wound healing steps is illustrated in Figure 1.

## 3. Galectin 1 in the Pathogenesis of Fibrosis

Non-resolving inflammation in different organs often leads to an accumulation of fibrotic tissue and allows over-production and deposition of ECM. The major cell type involved in this process are resident fibroblasts, which differentiate into active myofibroblast and starts to express Col-1 and α-smooth muscle actin (α-SMA). Epithelial–mesenchymal transition is another important source of myofibroblasts, as well as bone marrow-derived fibroblasts and pericytes. A key element on tissues repair and fibrosis are the macrophages, since they are the major source of transforming growth factor beta (TGF-β), the main pro-fibrotic cytokine which induces myofibroblast differentiation and collagen deposition.

### 3.1. Keloid Tissue

Skin keloids are dermal fibroproliferative tissue resulting from abnormal wound healing with pathophysiology not fully elucidated. Recently, the implication of Gal-1-glycan complexes in trans-differentiation of dermal fibroblasts into myofibroblasts and production of ECM components in keloid tissue was demonstrated. Interestingly, Gal-1 was overexpressed in the thickened epidermis and fibroblasts within the reticular dermis from the biopsies of patients diagnosed with keloid, suggesting its involvement in regulating the dermal fibroblast proliferation/dermal collagen production and contributing to the epidermal thickening and altered stratification/terminal keratinocyte differentiation [51]. The activated dermal fibroblasts and immune cells around abnormal microvasculature expressed versican, syndecan-1, fibronectin, thrombospondin-1, tenascin C, CD44, N-cadherin and integrin β1, while Gal-1 through their binding with ECM molecules formed a supramolecular structure at the cell surface of fibroblasts, immune cells and endothelial cells, and in the extracellular space that might influence the fibroblast phenotype and behavior related to adhesion, proliferation, migration and the inflammatory responses [52].

Since Gal-1 has a role in normal wound healing, the persistent upregulation of Gal-1 in keloid tissue suggests its contribution to angiogenesis and ECM production and fibrosis [52]. Over-expression of Gal-1 increases phosphorylation of Akt [53,54] and further Akt signaling may increase its transcription in a positive feedback response [55]. The PI3K/Akt pathway activation is a key regulator of myofibroblast differentiation and ECM production [56]. Interestingly, recent results showed a chronic increase in mRNA expression of Gal-1 in hypertrophic skin scars years later after wound healing, suggesting its role in the development of myofibroblast-induced collagen secretion and dermal thickening [57].

### 3.2. Diabetic Retinopathy

As a consequence of persistent inflammation or hypoxia, proliferative diabetic retinopathy (PDR) develops subsequently fibrovascular proliferative tissue on the retinal surface or into the vitreous cavity [58,59], resulting in retinal detachment [1]. Although it was noticed that Müller cells produce stress fibers that may provide mechanical forces for the retinal detachment process [60]. Under hypoxia conditions, the ocular microenvironment of the PDR patients produces many angiogenic factors, such as VEGF, promoting retinal neovascularization and vascular leakages [61,62]. During the PDR progression, the leukocytes emigrated into fibrovascular tissue and express pro-inflammatory and pro-angiogenic molecules [63,64,65]. Nevertheless, the presence of Gal-1 in leukocytes residents to epiretinal fibrovascular membranes of the PDR patients was recently demonstrated, while exposure to Müller cells induced VEGF upregulation and increased leukocyte adhesion to human retinal microvascular endothelial cells [66], suggesting its role in inflammation and neovascularization.

During hypoxic conditions, Müller cells overexpressed VEGF-A and contributed to the angiogenic promotion of diabetic retinopathy (DR) [67]. As a response to hypoxia, hypoxia-inducible factor 1-alpha (HIF-1α) is significantly up-regulated by PI3K/AKT and MAPK/ERK signaling [68,69]. Interestingly, HIF-1α and Gal-1 were found to be co-localized in Müller cells into epiretinal fibrovascular tissues of PDR patients, suggesting a significant contribution of HIF-1α to the expression of this lectin in glial cells [70].

Diabetic patients with pre-ischemia or inflammation and macular edema were correlated with Gal-1 eye overexpression [71]. In addition, the fibrovascular tissues from PDR accumulate advanced glycation endproducts (AGEs) and may activate interleukin 1 beta (IL-1β)-related inflammatory pathways in macrophages, followed by Müller cells, linking to Gal-1 upregulation in human DR progression [71].

### 3.3. Diabetic Nephropathy

Renal tubulointerstitial fibrosis is a major pathologic consequence of diabetic nephropathy (DN) which leads to renal failure [72,73]. Chronic exposure to hyperglycemia affects especially tubular cells and contributes to the tubulointerstitial changes [74,75].

Gal-1 is up-regulated in the kidneys of type 1 and 2 diabetic mice and in renal tubular cells exposed to hyperglycemia by p-Akt/AP4 transcriptional signaling, suggesting that lectin accumulation into the kidney cortex has a possible role in kidney fibrogenesis in diabetic mice [55]. In addition, in hyperglycemic conditions, human podocytes overexpressed Gal-1 and induced loss of podocin, suggesting that by interfering with podocin expression, Gal-1 may serve as a marker in diabetic nephropathy progression [76].

Clinical results showed that the higher serum Gal-1 levels of the patients with a higher incidence of hypertension, diabetes, chronic kidney disease, heart failure and multiple blood vessel dysfunctions, were found to be associated independently with a greater risk of renal function decline [77]. Moreover, Gal-1 was found to be increased in subcutaneous dialysates from type 2 diabetes compared with samples of healthy individuals [78].

### 3.4. Liver Fibrosis

Liver fibrosis is a response to the injury characterized by the accumulation of the abnormal extracellular matrix components, mainly due to the hepatic stellate cell (HSC) activation, which transdifferentiates from the quiescent form into activated myofibroblasts [79]. The activation and proliferation of HSC induced positive feedback of the pro-fibrotic pathways, as (TGF-β, which stimulates gene expression in activated myofibroblasts [80].

During HSCs trans-differentiation process into myofibroblasts and fibrosis progression, a significantly up-regulated expression of Gal-1, via the MEK1/2-ERK1/2 signaling pathway, was noticed [81]. In addition to HSC’s trans-differentiation and proliferation, Gal-1 promotes also HSCs migration; the exposure of LX2-cells to recombinant Gal-1 protein induced the phosphorylation of Smad-2,-3 and ERK1/2 and bind neuropilin 1 (NRP-1) in a glycosylation-dependent manner to enhance HSCs migration [81]. Other studies confirmed that fibrotic livers and activated HSCs overexpress Gal-1 and also induced NRP-1 N-glycosylation, which subsequently form a complex with PDGFRs and TGF-βRs in HSCs [82,83]; this complex further regulates Gal-1-induced HSC activation and migration [84]. In addition, blocking of endogenous Gal-1 expression suppressed PDGF- and TGF-β1-induced signaling, migration and mRNA expression in HSCs [84]. Other studies confirmed that Gal-1 gene expression silencing downregulated TGF-β1, connective tissue growth factor (CTGF) and α-SMA in HSCs and alleviates liver fibrosis in mice [85]. Additionally, silencing the mRNA expression of Gal-1 inhibited cell cycle progression, proliferation and migration and induced the apoptosis of HSCs from fibrotic livers in mice [85].

The role of Gal-1 in the activation and migration of HSCs is represented in Figure 2.

### 3.5. Pancreatic Fibrosis

After an injury or consecutive intense inflammation, quiescent pancreatic stellate cells are activated into myofibroblast-like cells, recognized by the increased expression of α-SMA, and produced extracellular matrix components, including Col-I [86,87].

There are few studies reporting Gal-1 expression in the pancreas, with some contradictory results [88,89]. In normal conditions, Gal-1 gene and protein expressions are at a low level [89,90], whereas it was higher expressed in fibroblasts of chronic pancreatitis samples [89] or in the extracellular matrix cells around the pancreatic cancer mass [90]. Other studies reported a strong expression of Gal-1 in the stroma surrounding the tumor, but negative in samples of chronic pancreatitis [88]. Gal-1, produced by activated pancreatic stellate cells, induced the recruitment of inflammatory cells by proliferation and production of monocyte chemoattractant protein-1 (MCP-1) and cytokine-induced neutrophil chemoattractant-1 (CINC-1), through activation of ERK, nuclear factor kappa-light-chain-enhancer of activated B cells (NF-kB) and in part by JNK and ERK pathways; the effects were abolished in the presence of thiodigalactoside, an inhibitor of Gal-1-galactoside binding [91]. Further, the recruitment of the inflammatory cells will increase the production and secretion of cytokines and growth factors, leading to support the pancreatic inflammation and for progression of pancreatic fibrosis.

### 3.6. Pulmonary Fibrosis

Idiopathic pulmonary fibrosis (IPF) is a disease caused by an accumulation of ECM proteins and trans-differentiation of lung fibroblasts to collagen-secreting myofibroblasts. The morphologic diagnosis of the fibrotic lungs is the presence of fibroblastic foci surrounded by hyperplastic type II alveolar epithelial cells [92], which act further in hypoxic conditions and over-express HIF-1α [93]. Other results, show that hypoxia signaling is an important factor in IPF progression [94], due to alveolar epithelial cells which can induce an increased production of TGF-β1 [95,96,97]. Profibrotic signaling pathways such as Wnt/β-catenin and TGF-β influence aberrant alveolar epithelial repair and fibrotic deposition [98,99]. In hypoxic conditions, these activities could be highlighted by different factors, such as focal adhesion kinase-1 (FAK1) which is involved in the trans-differentiation of fibroblasts into myofibroblasts [100]. Under these conditions, Gal-1 has been shown to act directly in exacerbating profibrotic signaling pathways and interacted with and activated FAK1 in lung epithelial cells. Contrarily, Gal-1 inhibition reduced FAK1 activation, preventing lung hypoxia and attenuating fibrosis progression [101].

The summary of the Gal-1 involvement in fibrosis pathogenesis is included in Table 1.

## 4. Conclusions and Future Prospects for Therapeutical Applications

In the first part, we described recent advances in the Gal-1 modulatory role on wound healing physiological process vs. skin scarring. By acting either in soluble or immobilized form, these glycan-binding proteins trigger different phases of tissue repair: hemostasis (platelet activation and aggregation via α_IIb_β3-integrin receptors); inflammation (neutrophil anti-trafficking and apoptosis, macrophage reprogramming and resolution of inflammation); proliferation, tissue repair and re-epithelialization (myofibroblast trans-differentiation via neuropilin-1/Smad3 signaling); when inflammation it is still present, the persistent upregulation of Gal-1 enhanced angiogenesis and ECM production via PI3K/Akt pathway activation and leads to keloid tissue. Since Gal-1 seems to play important roles in many wound healing stages, the targeted Gal-1 modulation should be considered as a method of choice for the treatment of wound healing, and to avoid keloid formation.

As discussed in the second part of the review article, previous studies have clarified the role of Gal-1 in the pathogenesis of keloid, proliferative diabetic retinopathy, liver fibrosis, renal fibrosis, pancreatic fibrosis, idiopathic pulmonary fibrosis and have suggested that Gal-1 may become a biomarker for the diagnosis and prognosis of tissue fibrosis in different diseases and potential therapeutic targets for its treatment. Inhibition of Gal-1 expression in dermal fibroblasts (keloid), Müller cells (diabetic retinopathy), renal epithelial cells (diabetic nephropathy), hepatic stellate cells (liver fibrosis), pancreatic hepatic cells (pancreatic fibrosis) and epithelial alveolar cells (pulmonary fibrosis) is what should be developed for therapeutic applications against organ fibrosis progression. Particularly, selective Gal-1 inhibitor OTX008 has been widely studied in pre-clinical in vitro and in vivo settings, where it showed a good Gal-1 inhibition activity [49,102,103]. This was not associated to apparent toxicity when intravenously injected in mice at the dose of 5 mg/kg [103]. Moreover, the only one ongoing clinical trial on OTX008 did not produced results yet, therefore a safety profile on this compound is not available yet in humans [104]. Overall, Gal-1 is thus a promising molecular target for the development of new and original therapeutic tools to treat fibrosis in different chronic diseases.

## Figures and Tables

**Figure 1 ijms-23-05548-f001:**
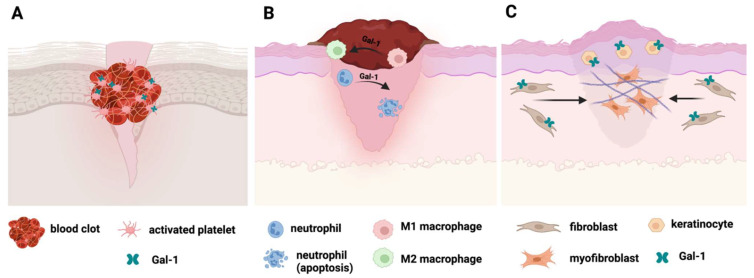
The role of Galectin 1 in wound healing. (**A**). Hemostasis; (**B**). Resolution of inflammation; (**C**). Proliferation, tissue repair and re-epithelization. Figure created with BioRender.com.

**Figure 2 ijms-23-05548-f002:**
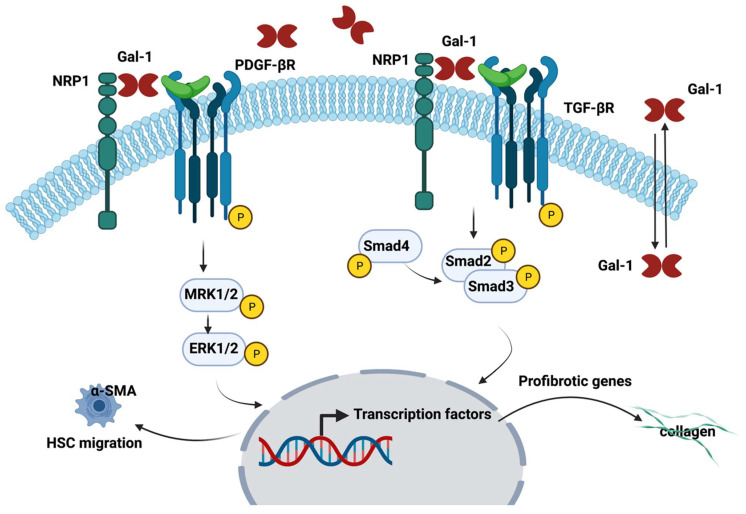
The role of Gal-1 in the HSCs activation and migration. Figure created with BioRender.com.

**Table 1 ijms-23-05548-t001:** The role of Gal-1 in the pathogenesis of fibrosis.

Disease	Experimental Models	Cells/Tissue Expressing Gal-1	Pathologic Effects	References
Keloid	Keloid patients	Fibroblasts localized in the papillary and reticular dermis	- Gal-1 overexpression in the thickened epidermis and dermal fibroblasts - Gal-1 induces fibroblasts trans-differentiation and ECM production	[51,52]
Proliferative diabetic retinopathy (PDR)	Human retinal Müller glial cells (MIOM1) streptozotocin-induced diabetic mice epiretinal fibrovascular membranes of PDR patients	Müller cells endothelial cells myofibroblasts leukocytes	- Retinal Gal-1 protein levels gradually increased over time in diabetic mice - Significant positive correlation between microvessel density, VEGF expression and the number of retinal blood vessels expressing Gal-1 in epiretinal - Fibrovascular membranes of PDR patients - Hypoxia induces overexpression of Gal-1 and HIF-1 in Müller glial cells, diabetic mice and PDR patients - Advanced glycation endproducts (AGEs) upregulate Gal-1 expression in Müller glial cells (in vitro and in vivo)	[66,70,71]
Liver fibrosis	LX2-cells TAA-induced liver fibrosis in mice CCl4- induced liver fibrosis in mice Gal-1 null mice	Hepatic stellate cells (HSC)	- Overexpression of Gal-1, via the MEK1/2-ERK1/2 signaling pathway - NRP-1/Gal-1/PDGFRs and TGF-βRs complex induce HSC activation and migration - Gal-1 gene expression silencing downregulates transforming growth factor (TGF-β1), connective tissue growth factor (CTGF) and α-smooth muscle actin (α-SMA) in HSCs and alleviates liver fibrosis in mice	[81,84,85]
Pancreatic fibrosis	Primary rat pancreatic stellate cells (PSCs)	Activated pancreatic fibroblasts (PSCs)	- Gal-1 induced the recruitment of inflammatory cells by proliferation and production of monocyte chemoattractant protein-1 (MCP-1) and cytokine-induced neutrophil chemoattractant-1 (CINC-1), through activation of ERK, NF-kB and in part by JNK and ERK pathways	[91]
Pulmonary fibrosis	H441 lung epithelial cells Primary mouse AEC cells	Lung epithelial cells	- Gal-1 promotes profibrotic signaling pathways and activates FAK1 in lung epithelial cells - Gal-1 inhibition reduced FAK1 activation, preventing lung hypoxia and attenuating fibrosis progression	[101]

## Data Availability

Not applicable.

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
