# Peer review of "Galectin 1—A Key Player between Tissue Repair and Fibrosis"

_ijms, 2022, doi:10.3390/ijms23105548_

Round 1

Reviewer 1 Report

Dear authors,

I liked a lot your review about galectin-1 role in tissue repair and fibrosis. However I have some comments and suggestions:

-Section 2.2: I miss a better explanation about how galectin 1 is involved in macrophages reprogramming and neutrophils. Please, explain a bit better how gal-1 plays a role, which models were used in the papers you are citing (in vivo or in vitro) and which molecular pathways and key genes/proteins are involved. The same applies to other sections, please, explain a bit more in which models and how they tested or showed the effects that you are describing, not only cite the article.

-Figures: I thank the authors for the effort they did illustrating each section with a figure, which is really nice. But, I feel the figures are very simple. Please, complete the figures with the key genes and pathways that are involved in each proces, also you need to include if galectin 1 is activating or inhibiting these molecular pathways (using arrows, or green/red colours...). For example in section 3.3 you talk about TGFb and PDGF, however any of these both molecules are mentioned in figure 5. 

Author Response

The authors would like to thank this Reviewer for appreciating the novelty of the paper and for supporting their conclusions. The answers to the criticism of this Reviewer are below.

-Section 2.2: I miss a better explanation about how galectin 1 is involved in macrophages reprogramming and neutrophils. Please, explain a bit better how gal-1 plays a role, which models were used in the papers you are citing (in vivo or in vitro) and which molecular pathways and key genes/proteins are involved. The same applies to other sections, please, explain a bit more in which models and how they tested or showed the effects that you are describing, not only cite the article.

-Figures: I thank the authors for the effort they did illustrating each section with a figure, which is really nice. But, I feel the figures are very simple. Please, complete the figures with the key genes and pathways that are involved in each proces, also you need to include if galectin 1 is activating or inhibiting these molecular pathways (using arrows, or green/red colours...). For example in section 3.3 you talk about TGFb and PDGF, however any of these both molecules are mentioned in figure 5. 

Response: Thank you for your suggestions which were very helpful in improving our review.  We reorganized the first part and the second. We introduced a subsection for keloid in the second part and finally, we summarized the 2.2. section in a table, the pro-fibrotic effects of Gal-1/disease, the experimental models used and the cells/tissues which overexpressed this protein. Therefore, we have left only the figure for the mechanisms by which gal-1 promotes HSC activation and migration in liver fibrosis, which was completely changed.

Reviewer 2 Report

Dear authors,

The review gives a nice overview of the association of Gal-1 with fibrogenesis. The scientific character is good and gives a lot of leads for further research.

That said, I have a few minor comments.

1: Check the English grammar. This distracts from the massage.

2: In the liver fibrosis part you referred to the role of Gal-1 in migration and activation of HSC. You might want to explain these mechanisms in more depth. 

3: In the conclusion you suggested targeted Gal-1 treatments. Can you explain whether there are Gal-1 targeted treatments already in use or in clinical trials? Can you explain something about safety. If this safety profile is not known for humans you might discus the safety in animal models?

Author Response

The authors would like to thank this Reviewer for appreciating the novelty of the paper and for supporting their conclusions. The answers to the criticism of this Reviewer are below.

1: Check the English grammar. This distracts from the massage.

Response: Indeed there were some mistakes. We corrected the manuscript

2: In the liver fibrosis part you referred to the role of Gal-1 in migration and activation of HSC. You might want to explain these mechanisms in more depth. 

Response: We extended the text (section 3.4) and we changed the figure which explains the mechanisms better (figure 2)

3: In the conclusion you suggested targeted Gal-1 treatments. Can you explain whether there are Gal-1 targeted treatments already in use or in clinical trials? Can you explain something about safety. If this safety profile is not known for humans you might discus the safety in animal models?

Response: Thanks for the suggestion. We mentioned the only clinical trial testing OTX008 available from literature (reference 106) in the Conclusion section, but unfortunatelly there are not data descripted about safety and toxicity for this study, as you can check at https://clinicaltrials.gov/ct2/show/study/NCT01724320. We also added in the same section a description of preclinical evidences using OTX008, with a particular description of OTX008 safety profile reported by an in vivo study (section 4).

Reviewer 3 Report

Hermenean et al review the role of Gal1 as modulator of tissue healing and dysfunction. The review points out a few important aspects on how galectins can modulate cellular and molecular processes and pathways, however, considering that Gal1 is affecting a variety of different important pathways of cell homeostasis, the review tackles the topic sometimes a bit superficially and sometimes in part 2 selects diseases randomly at least from a readers' perspective without a remark on the motivation or selection criteria.

Remarks:

  • Language and synthax editing is required e.g. tissues repair, ...fibroblasts starts to express..., wound hiling, Gal-1 eye overexoression, etc.
  • unclear why a sentence of figure callout needs a paragraphing
  • Part 1 sometimes mentions other Gals even though the review focuses on Gal1 - seems unnecessary
  • Part 2 on pathogenesis: Why did the authors decide to focus on tgf, hypoxia etc. on just a handful of tissues that can be fibrotic. This seems very selective considering part1 is focusing skin wound healing; 
  • part 2 selection on fibrotic tissues seems ambiguous as no explanation if given why i.e. keloids are not mentioned which is also strong case of fibroblast hyperactivity thus fibrosis

Author Response

The authors would like to thank this Reviewer for appreciating the novelty of the paper and for supporting their conclusions. The answers to the criticism of this Reviewer are below.

  1. Language and synthax editing is required e.g. tissues repair, ...fibroblasts starts to express..., wound hiling, Gal-1 eye overexoression, etc.

Response: We corrected the manuscript

  1. Unclear why a sentence of figure callout needs a paragraphing

Response: We rephrased or erased some paragraphs

  1. Part 1 sometimes mentions other Gals even though the review focuses on Gal1 - seems unnecessary

Response:We eliminated as your sugestion

  1. Part 2 on pathogenesis: Why did the authors decide to focus on tgf, hypoxia etc. on just a handful of tissues that can be fibrotic. This seems very selective considering part1 is focusing skin wound healing;

part 2 selection on fibrotic tissues seems ambiguous as no explanation if given why i.e. keloids are not mentioned which is also strong case of fibroblast hyperactivity thus fibrosis

Response: Thank you for your suggestion. We reorganized the first part and the second. We introduced a subsection for keloid in the second part and finally, we summarized the effects in a table. Therefore, we have left only the figure for the mechanisms by which gal-1 promotes HSC activation and migration in liver fibrosis.

Round 2

Reviewer 3 Report

Authors revised their manuscript accordingly